# LexMAE: Lexicon-Bottlenecked Pretraining for Large-Scale Retrieval

**Tao Shen, Xiubo Geng, Chongyang Tao, Can Xu, Xiaolong Huang, Binxing Jiao, Linjun Yang, Daxin Jiang***

Microsoft. {shentao,xigeng,chotao,caxu,xiaolhu, binxjia,linjya,djiang}@microsoft.com

## Abstract

In large-scale retrieval, the lexicon-weighting paradigm, learning weighted sparse representations in vocabulary space, has shown promising results with high quality and low latency. Despite it deeply exploiting the lexicon-representing capability of pre-trained language models, a crucial gap remains between language modeling and lexicon-weighting retrieval – the former preferring certain or low-entropy words whereas the latter favoring pivot or high-entropy words – becoming the main barrier to lexicon-weighting performance for large-scale retrieval. To bridge this gap, we propose a brand-new pre-training framework, lexicon-bottlenecked masked autoencoder (LexMAE), to learn importance-aware lexicon representations. Essentially, we present a lexicon-bottlenecked module between a normal language modeling encoder and a weakened decoder, where a continuous bag-of-words bottleneck is constructed to learn a lexicon-importance distribution in an unsupervised fashion. The pre-trained LexMAE is readily transferred to the lexicon-weighting retrieval via fine-tuning. On the ad-hoc retrieval benchmark, MS-Marco, it achieves 42.6% MRR@10 with 45.8 QPS for the passage dataset and 44.4% MRR@100 with 134.8 QPS for the document dataset, by a CPU machine. And LexMAE shows state-of-the-art zero-shot transfer capability on BEIR benchmark with 12 datasets.

## 1 Introduction

Large-scale retrieval, also known as first stage retrieval (Cai et al., 2021), aims to fetch top *query*-relevant documents from a huge collection. In addition to its indispensable roles in dialogue systems (Zhao et al., 2020), question answering (Karpukhin et al., 2020), search engines, etc., it also has been surging in recent cutting-edge topics, e.g., retrieval-augmented generation (Lewis et al., 2020) and retrieval-augmented language modeling (Guu et al., 2020). As there are millions to billions of documents in a collection, efficiency is the most fundamental prerequisite for large-scale retrieval. To this end, query-agnostic document representations (i.e., indexing the collection independently) and lightweight relevance metrics (e.g., cosine similarity, dot-product) have become the common practices to meet the prerequisite – usually achieved by a two-tower structure (Reimers & Gurevych, 2019), a.k.a., bi-encoder and dual-encoder, in representation learning literature.

Besides the prevalent 'dense-vector retrieval' paradigm that encodes both queries and documents in the same low-dimension, real-valued latent semantic space (Karpukhin et al., 2020), another retrieval paradigm, 'lexicon-weighting retrieval', aims to leverage weighted sparse representation in vocabulary space (Formal et al., 2021a; Shen et al., 2022). It learns to use a few lexicons in the vocabulary and assign them with weights to represent queries and documents – sharing a high-level inspiration with BM25 but differing in dynamic (with compression and expansion) lexicons and their importance weights learned in an end-to-end manner. Although learning the representations in such a high-dimensional vocabulary space seems intractable with limited human-annotated query-document pairs, recent surging pre-trained language modeling (PLM), especially the masked language modeling (MLM), facilitates transferring context-aware lexicon-coordinate knowledge into lexicon-weighting retrieval by fine-tuning the PLM on the annotated pairs (Formal et al., 2021b;a; Shen et al., 2022). Here, coordinate terms (full of synonyms and concepts) are highly related to relevance-centric tasks and mitigate the lexicon mismatch problem (Cai et al., 2021), leading to superior retrieval quality.

---

*Corresponding author.

Due to the pretraining-finetuning consistency with the same output vocabulary space, lexicon-based retrieval methods can fully leverage a PLM, including its masked language modeling (MLM) head, leading to better search quality (e.g., $\sim 1.0\%$ MRR@10 improvement over dense-vector ones by fine-tuning the same PLM initialization (Formal et al., 2021a; Hofstätter et al., 2020)). Meantime, attributed to the high-dimensional sparse-controllable representations (Yang et al., 2021; Lassance & Clinchant, 2022), these methods usually enjoy higher retrieval efficiency than dense-vector ones (e.g., $10\times$ faster with the identical performance in our experiments).

Nonetheless, there still exists a subtle yet crucial gap between the pre-training language modeling and the downstream lexicon-weighting objectives. That is, MLM (Devlin et al., 2019) aims to recover a word back given its contexts, so it inclines to assign high scores to certain (i.e., low-entropy) words, but these words are most likely to be articles, prepositions, etc., or belong to collocations or common phrases. Therefore, language modeling is in conflict with the lexicon-weighting representation for relevance purposes, where the latter focuses more on the high-entropy words (e.g., subject, predicate, object, modifiers) that are essential to the semantics of a query or document. These can explain, in our experiments when being fine-tuned under the paradigm of lexicon-weighting retrieval (Formal et al., 2021a), why a moderate PLM (i.e., DistilBERT) can even outperform a relatively large one (i.e., BERT-base) and why a well-trained PLM (e.g., RoBERTa) cannot even achieve a convergence.

To mitigate the gap, in this work we propose a brand-new pre-training framework, dubbed lexicon-bottlenecked masked autoencoder (LexMAE), to learn importance-aware lexicon representations for the transferable knowledge towards large-scale lexicon-weighting retrieval. Basically, LexMAE pre-trains a language modeling encoder for document-specific lexicon-importance distributions over the whole vocabulary to reflect each lexicon's contribution to the document reconstruction. Motivated by recent dense bottleneck-enhanced pre-training (Gao & Callan, 2022; Liu & Shao, 2022; Wang et al., 2022), we present to learn the lexicon-importance distributions in an unsupervised fashion by constructing continuous bag-of-words (CBoW) bottlenecks upon the distributions.

Thereby, LexMAE pre-training architecture consists of three components: i) a language modeling encoder (as most other PLMs, e.g., BERT, RoBERTa), ii) a lexicon-bottlenecked module, and iii) a weakened masking-style decoder. Specifically, a mask-corrupted document from the collection is passed into the *language modeling encoder* to produce token-level LM logits in the vocabulary space. Besides an MLM objective for generic representation learning, a max-pooling followed by a normalization function is applied to the LM logits to derive a lexicon-importance distribution. To unsupervisedly learn such a distribution, the *lexicon-bottlenecked module* leverages it as the weights to produce a CBoW dense bottleneck, while the *weakened masking-style decoder* is asked to reconstruct the aggressively masked document from the bottleneck. Considering the shallow decoder and its aggressive masking, the decoder in LexMAE is prone to recover the masked tokens on the basis of the CBoW bottleneck, and thus the LexMAE encoder assigns higher importance scores to essential vocabulary lexicons of the masked document but lower to trivial ones. This closely aligns with the target of the lexicon-weighting retrieval paradigm and boosts its performance.

After pre-training LexMAE on large-scale collections, we fine-tune its language modeling encoder to get a lexicon-weighting retriever, improving previous state-of-the-art performance by 1.5% MRR@10 with $\sim 13\times$ speed-up on the ad-hoc passage retrieval benchmark. Meantime, LexMAE also delivers new state-of-the-art results (44.4% MRR@100) on the ad-hoc document retrieval benchmark. Lastly, LexMAE shows great zero-shot transfer capability and achieves state-of-the-art performance on BEIR benchmark with 12 datasets, e.g., Natural Questions, HotpotQA, and FEVER. [1]

## 2 RELATED WORK

**PLM-based Dense-vector Retrieval.** Recently, pre-trained language models (PLM), e.g., BERT (Devlin et al., 2019), RoBERTa (Liu et al., 2019), DeBERTa (He et al., 2021b), have been proven generic and effective when transferred to a broad spectrum of downstream tasks via fine-tuning. When transferring PLMs to large-scale retrieval, a ubiquitous paradigm is known as 'dense-vector retrieval' (Xiong et al., 2021) – encoding both queries and documents in the same low-dimension semantic space and then calculating query-document relevance scores on the basis of spatial distance. However, dense-vector retrieval methods suffer from the objective gap between lexicon-recovering

---

[1] We released our codes and models at `https://github.com/taoshen58/LexMAE`.

language model pre-training and document-compressing dense-vector fine-tuning. Although a natural remedy has been dedicated to the gap by constructing pseudo query-document pairs (Lee et al., 2019; Chang et al., 2020; Gao & Callan, 2022; Zhou et al., 2022a) or/and enhancing bottleneck dense representation (Lu et al., 2021; Gao & Callan, 2021; 2022; Wang et al., 2022; Liu & Shao, 2022), the methods are still limited by their intrinsic representing manners – dense-vector leading to large index size and high retrieval latency – applying speed-up algorithms, product-quantization (Zhan et al., 2022), however resulting in dramatic drops (e.g., $-3\% \sim 4\%$ by Xiao et al. (2022)).

**Lexicon-weighing Retrieval.**  In contrast to the almost unlearnable BM25, lexicon-weighing retrieval methods, operating on lexicon-weights by a neural model, are proposed to exploit language models for term-based retrieval (Nogueira et al., 2019b;a; Formal et al., 2021b;a; 2022). According to different types of language models, there are two lines of work: based on causal language models (CLM) (Radford et al.; Raffel et al., 2020), (Nogueira et al., 2019a) use the concurrence between a document and a query for lexicon-based sparse representation expansion. Meantime, based on masked language models (MLM) (Devlin et al., 2019; Liu et al., 2019), (Formal et al., 2021b) couple the original word with top coordinate terms (full of synonyms and concepts) from the pre-trained MLM head. However, these works directly fine-tune the pre-trained language models, regardless of the objective mismatch between general language modeling and relevance-oriented lexicon weighting.

## 3 LexMAE: Lexicon-bottlenecked Masked Autoencoder

**Overview of LexMAE Pre-training.**  As illustrated in Figure 1, our lexicon-bottlenecked masked autoencoder (LexMAE) contains one encoder and one decoder with masked inputs in line with the masked autoencoder (MAE) family (He et al., 2021a; Liu & Shao, 2022), while is equipped with a lexicon-bottlenecked module for document-specific lexicon-importance learning.

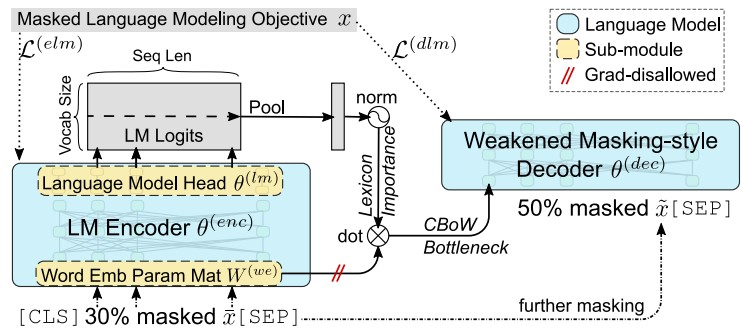

Figure 1: An illustration of lexicon-bottlenecked masked autoencoder (LexMAE) pre-training architecture.

Given a piece of free-form text, $x$, from a large-scale collection, $\mathbb{D}$, we aim to pre-train a language modeling encoder, $\theta^{(\text{enc})}$, that represents $x$ with weighted lexicons in the vocabulary space, i.e., $\boldsymbol{a} \in [0,1]^{|\mathbb{V}|}$. $\mathbb{V}$ denotes the whole vocabulary. Here, each $a_i = P(\text{w} = w_i | x; \theta^{(\text{enc})})$ with $w_i \in \mathbb{V}$ denotes the importance degree of the lexicon $w_i$ to the whole text $x$. To learn the distribution $\boldsymbol{a}$ for $x$ in an unsupervised fashion, an additional decoder $\theta^{(\text{dec})}$ is asked to reconstruct $x$ based on $\boldsymbol{a}$.

### 3.1 Language Modeling Encoder

Identical to most previous language modeling encoders, e.g., BERT (Devlin et al., 2019), the language modeling encoder, $\theta^{(\text{enc})}$, in LexMAE is composed of three parts, i.e., a *word embedding module* mapping the discrete tokens of $x$ to dense vectors, a *multi-layer Transformer* (Vaswani et al., 2017) for deep contextualization, and a *language modeling head* mapping back to vocabulary space $\mathbb{R}^{|\mathbb{V}|}$.

First, following the common practice of pre-training the encoder unsupervisedly, a masked language modeling (MLM) objective is employed to pre-train $\theta^{(\text{enc})}$. Formally, given a piece of text $x \in \mathbb{D}$, a certain percentage ($\alpha\%$) of the tokens in $x$ are masked to obtain $\bar{x}$, in which $80\%$ replaced with a special token [MASK], $10\%$ replaced with a random token in $\mathbb{V}$, and the remaining kept unchanged (Devlin et al., 2019). Then, the masked $\bar{x}$ is fed into the language modeling encoder, $\theta^{(\text{enc})}$, i.e.,

$$\boldsymbol{S}^{(\text{enc})} = \text{Transformer-LM}(\bar{x}; \theta^{(\text{enc})}) \in \mathbb{R}^{|\mathbb{V}| \times n}, \tag{1}$$

where $\boldsymbol{S}^{(\text{enc})}$ denotes LM logits. Lastly, the MLM objective is to minimize the following loss,

$$L^{(\text{elm})} = -\sum_{\mathbb{D}} \sum_{j \in \mathbb{M}^{(\text{enc})}} \log P(\text{w}^j = x_j | \bar{x}; \theta^{(\text{enc})}), \text{ where } P(\text{w}^j) := \text{softmax}(\boldsymbol{S}^{(\text{enc})}_{:,j}), \tag{2}$$

where $\mathbb{M}^{(\text{enc})}$ denotes the set of masked indices of the tokens in $\bar{x}$, $\text{w}^j$ denotes the discrete variable over $\mathbb{V}$ at the $j$-th position of $x$, and $x_j$ is its original token (i.e., golden label of the MLM objective).

## 3.2 LEXICON-BOTTLENECKED MODULE

Given token-level logits from Eq.(1) defined in $\mathbb{V}$, we calculate a lexicon-importance distribution by

$$\boldsymbol{a} := P(\text{w}|\bar{x}; \theta^{(\text{enc})}) = \text{Normalize}(\text{Max-Pool}(\boldsymbol{S}^{(\text{enc})})) \in [0, 1]^{|\mathbb{V}|}, \qquad (3)$$

where $\text{Max-Pool}(\cdot)$ is pooling along with its sequence axis, which is proven more effective than mean-pooling in lexicon representation (Formal et al., 2021a), $\text{Normalize}(\cdot)$ is a normalization function (let $\sum a_i = 1$), which we simply take $\text{softmax}(\cdot)$ in our main experiments. $P(\text{w}|\bar{x}; \theta^{(\text{enc})})$ is lexicon-importance distribution over $\mathbb{V}$ to indicate which lexicons in $\mathbb{V}$ is relatively important to $x$.

The main obstacle to learning the lexicon-importance distribution $P(\text{w}|\bar{x}; \theta^{(\text{enc})})$ is that we do not have any general-purpose supervised signals. Inspired by recent bottleneck-enhanced dense representation learning (Gao & Callan, 2022; Liu & Shao, 2022; Wang et al., 2022), we propose to leverage the lexicon-importance distribution as a clue for reconstructing $x$ back. As such, our language modeling encoder will be prone to focus more on the pivot or essential tokens/words in $x$. However, it is intractable to directly regard the high-dimensional distribution vector $\boldsymbol{a} \in [0, 1]^{|\mathbb{V}|}$ as a bottleneck since i) the distribution over the whole $\mathbb{V}$ has enough capacity to hold most semantics of $x$ (Yang et al., 2018), making the bottleneck less effective, and ii) the high-dimensional vector is hardly fed into a decoder for representation learning and text reconstruction.

Therefore, we further propose to construct a continuous bag-of-words (CBoW) bottleneck following the lexicon-importance distribution $P(\text{w}|\bar{x}; \theta^{(\text{enc})})$ derived from Eq.(3). That is

$$\boldsymbol{b} := \mathbb{E}_{w_i \sim P(\text{w}|\bar{x}; \theta^{(\text{enc})})}[\boldsymbol{e}^{(w_i)}] = \boldsymbol{W}^{(\text{we})}\boldsymbol{a}. \qquad (4)$$

Here, $\boldsymbol{W}^{(\text{we})} = [\boldsymbol{e}^{(w_1)}, \boldsymbol{e}^{(w_2)}, \dots] \in \mathbb{R}^{d \times |\mathbb{V}|}$ denotes the learnable word embedding matrix in the parameters $\theta^{(\text{enc})}$ of language modeling encoder, where $d$ denotes embedding size, $\boldsymbol{e}^{(w_i)} \in \mathbb{R}^d$ is a word embedding of the lexicon $w_i$. Thereby, $\boldsymbol{b} \in \mathbb{R}^d$ stands for dense-vector CBoW bottleneck, upon which a decoder (will be detailed in the next sub-section) is asked to reconstruct the original $x$ back.

**Remark.** As aforementioned in our Introduction, there exists a conflict between MLM and lexicon-importance objectives, but we still apply an MLM objective in our encoder. This is because i) the MLM objective can serve as a regularization term to ensure the original token in $x$ receive relatively high scores in contrast to its coordinate terms and ii) the token-level noise introduced by the MLM task has been proven effective in robust learning.

## 3.3 WEAKENED MASKING-STYLE DECODER

Lastly, to instruct the bottleneck representation $\boldsymbol{b}$ and consequently learn the lexicon-importance distribution $P(\text{w}|\bar{x}; \theta^{(\text{enc})})$, we leverage a decoder to reconstruct $x$ upon $\boldsymbol{b}$. In line with recent bottleneck-enhanced neural structures (Gao & Callan, 2022; Wang et al., 2022), we employ a weakened masking-style decoder parameterized by $\theta^{(\text{dec})}$, which pushes the decoder to rely heavily on the bottleneck representation. It is noteworthy that the 'weakened' is reflected by two-fold: i) aggressively masking strategy and ii) shallow Transformer layers (says two layers).

In particular, given the masked input at the encoder side, $\bar{x}$, we first apply an extra $\beta\%$ masking operation, resulting in $\tilde{x}$. That is, the decoder is required to recover all the masked tokens that are also absent in the encoder, which prompts the encoder to compress rich contextual information into the bottleneck. Then, we prefix $\tilde{x}$ with the bottleneck representation $b$, i.e., replacing the special token `[CLS]` with the bottleneck. Therefore, our weakened masking-style decoding with a Transformer-based language modeling decoder can be formulated as

$$\boldsymbol{S}^{(\text{dec})} = \text{Transformer-LM}(\boldsymbol{b}, \tilde{x}; \theta^{(\text{dec})}) \in \mathbb{R}^{|\mathbb{V}| \times n}, \qquad (5)$$

where $\theta^{(\text{dec})}$ parameterizes this weakened masking-style decoder. Lastly, similar to the MLM at the encoder side, the loss function is defined as

$$L^{(\text{dlm})} = -\sum_{\mathbb{D}} \sum_{j \in \mathbb{M}^{(\text{dec})}} \log P(\text{w}^j = x_j | \tilde{x}; \theta^{(\text{dec})}), \text{ where } P(\text{w}^j) := \text{softmax}(\boldsymbol{S}^{(\text{dec})}_{:,j}), \qquad (6)$$

where $\mathbb{M}^{(\text{dec})}$ denotes the set of masked indices of the tokens in the decoder's input, $\tilde{x}$.

### 3.4 Pre-Training Objective & Fine-tuning for Lexicon-weighting Retriever

The final loss of pre-training LexMAE is an addition of the losses defined in Eq.(2) and Eq.(6), i.e.,

$$L^{(\text{lm})} = L^{(\text{elm})} + L^{(\text{dlm})}. \tag{7}$$

Meanwhile, we tie all word embedding metrics in our LexMAE pre-training architecture, including the word embedding modules and language model heads of both the encoder & decoder, as well as $\boldsymbol{W}^{(\text{we})}$ in Eq.(4). It is noteworthy that we cut-off the gradient back-propagation for $\boldsymbol{W}^{(\text{we})}$ in Eq.(4) to make the training focus only on the lexicon-importance distribution $P(\text{w}|\bar{x}; \theta^{(\text{enc})})$ rather than $\boldsymbol{W}^{(\text{we})}$.

**Task Definition of Downstream Large-scale Retrieval.**   Given a collection containing a number of documents, i.e., $\mathbb{D} = \{d_i\}_{i=1}^{|\mathbb{D}|}$, and a query $q$, a retriever aims to fetch a list of text pieces $\bar{\mathbb{D}}_q$ to contain all relevant ones. Generally, this is based on a relevance score between $q$ and every document $d_i$ in a Siamese manner, i.e., $< \text{Enc}(q), \text{Enc}(d_i) >$, where $\text{Enc}$ is an arbitrary representation model (e.g., neural encoders) and $< \cdot, \cdot >$ denotes a lightweight relevance metric (e.g., dot-product).

To transfer LexMAE into large-scale retrieval, we get rid of its decoder but only fine-tune the language modeling encoder for the lexicon-weighting retriever. Basically, to leverage a language modeling encoder for lexicon-weighting representations, we adopt (Formal et al., 2021a) and represent a piece of text, $x$, in high-dimensional vocabulary space by

$$\boldsymbol{v}^{(x)} = \log(1 + \text{Max-Pool}(\max(\text{Transformer-LM}(x; \theta^{(\text{enc})}), 0))) \in \mathbb{R}*^{|\mathbb{V}|}, \tag{8}$$

where $\max(\cdot, 0)$ ensures all values greater than or equal to zero for upcoming sparse requirements, and the saturation function $\log(1 + \text{Max-Pool}(\cdot))$ prevents some terms from dominating.

In contrast to a classification task, the retrieval tasks are formulated as contrastive learning problems. That is, only a limited number of positive documents, $d_+^{(q)}$, is provided for a query $q$ so we need sample a set of negative documents, $\mathbb{N}^{(q)} = \{d_-^{(q)}, \dots\}$, from $\mathbb{D}$ for the $q$. And we will dive into various sampling strategies to get $\mathbb{N}^{(q)}$ in §A. Note that if no confusion arises, we omit the superscript $(q)$ that indicates 'a query-specific' for clear demonstrations. By following Shen et al. (2022), we can first derive a likelihood distribution over the positive $\{d_+\}$ and negative $\mathbb{N}$ documents, i.e.,

$$\boldsymbol{p} := P(\text{d}|q, \{d_+\} \cup \mathbb{N}; \theta^{(\text{enc})}) = \frac{\exp(\boldsymbol{v}^{(q)T}\boldsymbol{v}^{(d)})}{\sum_{d' \in \{d_+\} \cup \mathbb{N}} \exp(\boldsymbol{v}^{(q)T}\boldsymbol{v}^{(d')})}, \, \forall d \in \{d_+\} \cup \mathbb{N} \tag{9}$$

where $\boldsymbol{v}^{(\cdot)} \in \mathbb{R}*^{|\mathbb{V}|}$ derived from Eq.(8) denotes a lexicon-weighting representation for a query $q$ or a document $d$. Then, the loss function of the contrastive learning towards this retrieval task is defined as

$$L^{(\text{r})} = \sum_q -\log P(\text{d} = d_+|q, \{d_+\} \cup \mathbb{N}; \theta^{(\text{enc})}) + \lambda \, \text{FLOPS}(q, d) = \sum -\log \boldsymbol{p}_{[\text{d}=d_+]} + \lambda \, \text{FLOPS}, \tag{10}$$

where $\text{FLOPS}(\cdot)$ denotes a regularization term for representation sparsity (Paria et al., 2020) as first introduced by Formal et al. (2021b) and $\lambda$ denotes a hyperparameter of it loss weight. Note that, to train a competitive retriever, we adapt the fine-tuning pipeline in (Wang et al., 2022), which consists of three stages (please refer to §A & §B for our training pipeline and inference details).

**Top-K Sparsifying.**   Attributed to inherent flexibility, we can adjust the sparsity of the lexicon-weighting representations for the documents to achieve a targeted efficacy-efficiency trade-off. Here, the 'sparsity' denotes how many lexicons in the vocabulary we used to represent each document. Previous methods either tune sparse regularization strength (Formal et al., 2021a; 2022) (e.g., $\lambda$ in Eq.(10)) or propose other sparse hyperparameters (Yang et al., 2021; Lassance & Clinchant, 2022) (e.g., the number of activated lexicons), however causing heavy fine-tuning overheads. Hence, we present a simple but effective sparsifying method, which only presents during embedding documents in the inference phase so requires almost zero extra overheads. It only keeps top-K weighted lexicons in the representations $\boldsymbol{v}^{(d)} \in \mathbb{R}*^{|\mathbb{V}|}$ by Eq.(8), while removing the others by assigning zero weights (see §D for details). We will dive into empirical efficacy-efficiency analyses later in §4.2.

## 4 Experiment

**Benchmark Datasets.**   Following Formal et al. (2021a), we first employ the widely-used passage retrieval datasets, MS-Marco (Nguyen et al., 2016). We only leverage its official queries (no

Table 1: Passage retrieval results on MS-Marco Dev, TREC Deep Learning 2019 (DL'19), and TREC Deep Learning 2020 (DL'20). M@10 and nDCG denotes MRR@10 and nDCG@10, respectively. The 'coCon' denotes the coCondenser that continually pre-trained BERT in an unsupervised manner, and the subscript of a pre-trained model denotes its scale (e.g., 'base' equal to 110M parameters). †Please refer to Table 8.

| Method | Pre-trained model | Reranker distilled | Hard negs | Multi Vec | MS-Marco Dev | | | DL'19 nDCG | DL'20 nDCG |
|---|---|---|---|---|---|---|---|---|---|
| | | | | | M@10 | R@100 | R@1k | | |
| *Dense-vector Retriever* | | | | | | | | | |
| ANCE (Xiong et al., 2021) | $RoBERTa_b$ | | | | 33.8 | 86.2 | 96.0 | 65.4 | 64.6 |
| ADORE (Zhan et al., 2021) | $RoBERTa_b$ | | ✓ | | 34.7 | 87.6 | - | 68.3 | - |
| TAS-B (Hofstätter et al., 2021) | DistilBERT | ✓ | | | 34.7 | - | 97.8 | 71.2 | 69.3 |
| TCT-ColBERT (Lin et al., 2021) | $BERT_{base}$ | ✓ | ✓ | | 35.9 | - | 97.0 | 71.9 | - |
| coCondenser (Gao & Callan, 2022) | $coCon_{base}$ | | ✓ | | 38.2 | - | 98.4 | 71.7 | 68.4 |
| ColBERTv2 (Santhanam et al., 2021) | $BERT_{base}$ | ✓ | ✓ | ✓ | 39.7 | - | 98.4 | - | - |
| RocketQAv2 (Ren et al., 2021b) | $ERNIE_{base}$ | ✓ | ✓ | | 38.8 | - | -† | - | - |
| AR2 (Zhang et al., 2022) | $coCon_{base}$ | ✓ | ✓ | | 39.5 | - | -† | - | - |
| SimLM (Wang et al., 2022) | $SimLM_{base}$ | ✓ | ✓ | | 41.1 | - | 98.7 | 71.2 | 69.7 |
| *Lexicon-base or Sparse Retriever* | | | | | | | | | |
| BM25 (Dai & Callan, 2019) | - | | | | 18.5 | 58.5 | 85.7 | 51.2 | 47.7 |
| DeepCT (Dai & Callan, 2019) | $BERT_{base}$ | | | | 24.3 | - | 91.3 | 55.1 | - |
| RepCONC (Zhan et al., 2022) | $RoBERTa_b$ | | ✓ | | 34.0 | 86.4 | - | 66.8 | - |
| SPLADE-max (Formal et al., 2021a) | DistilBERT | | | | 34.0 | - | 96.5 | 68.4 | - |
| DistilSPLADE-max (Formal et al., 2021a) | DistilBERT | ✓ | | | 36.8 | - | 97.9 | 72.9 | - |
| SelfDistil (Formal et al., 2022) | DistilBERT | ✓ | ✓ | | 36.8 | - | 98.0 | 72.3 | - |
| Co-SelfDistil (Formal et al., 2022) | $coCon_{base}$ | ✓ | ✓ | | 37.5 | - | 98.4 | 73.0 | - |
| **LexMAE** | $LexMAE_{base}$ | ✓ | ✓ | | **42.6** | **93.1** | **98.8** | **73.7** | **72.8** |

augmentations (Ren et al., 2021b)), and report for MS-Marco Dev set, TREC Deep Learning 2019 set (Craswell et al., 2020), and TREC Deep Learning 2020 set (Craswell et al., 2021). Besides, we evaluate the zero-shot transferability of our model on BEIR benchmark (Thakur et al., 2021). We employ twelve datasets covering semantic relatedness and relevance-based retrieval tasks (i.e., TREC-COVID, NFCorpus, Natural Questions, HotpotQA, FiQA, ArguAna, Tóuche-2020, DBPedia, Scidocs, Fever, Climate-FEVER, and SciFact) in the BEIR benchmark as they are widely-used across most previous retrieval works. Lastly, to check if our LexMAE is also compatible with long-context retrieval, we conduct document retrieval evaluations on MS-Marco Doc Dev. Note that if not specified in our analyzing sections of the remainder, the numbers are reported on MS-Marco passage dev.

**Evaluation Metrics.** We report MRR@10 (M@10) and Recall@1/50/100/1K for MS-Marco Dev (passage), and report NDCG@10 for both TREC Deep Learning 2019 (passage) and TREC Deep Learning 2020 (passage). Moreover, NDCG@10 is reported on BEIR benchmark, while MRR@100 and Recall@100 are reported for MS-Marco Doc. Regarding R@N metric, we found there are two kinds of calculating ways, and we strictly follow the official evaluation (please refer to §C).

**Setups.** We pre-train on the MS-Marco collection (Nguyen et al., 2016), where most hyperparameters are identical to (Wang et al., 2022): the encoder is initialized by $BERT_{base}$ (Devlin et al., 2019) whereas the others are randomly initialized, the batch size is 2048, the max length is 144, the learning rate is $3 \times 10^{-4}$, the number of training steps is 80k, the masking percentage ($\alpha\%$) of encoder is 30%, and that ($\alpha + \beta\%$) of decoder is 50%. Meantime, the random seed is always 42, and the pre-training is completed on 8×A100 GPUs within 14h. Please refer to §A.2 for our fine-tuning setups.

## 4.1 MAIN EVALUATION

**MS-Marco Dev (Passage Retrieval).** First, we compare our fine-tuned LexMAE with a wide range of baselines and competitors to perform large-scale retrieval in Table 1. It is shown that our method substantially outperforms the previous best retriever, SimLM, by a very large margin (+1.5% MRR@10) and achieves a new state-of-the-art performance. Standing with different retrieval paradigms and thus different bottleneck constructions, such a large performance margin verifies the superiority of lexicon-weighting retrieval when a proper initialization is given. Meantime, the LexMAE is dramatically superior (+5.1% MRR@10) to its baseline (Formal et al., 2022), Co-Self-Disil, with the same neural model scale but different model initialization (coCondenser (Gao & Callan, 2022) v.s. LexMAE). This verifies that our lexicon-bottlenecked pre-training is more effective than the dense-bottlenecked one in lexicon-weighting retrieval.

Table 2: Zero-shot transfer performance (nDCG@10) on BEIR benchmark. 'BEST ON' and 'AVERAGE' do not take the in-domain result into account. 'ColBERT' is its v2 version (Santhanam et al., 2021).

| Method | BM25 | DocT5 | SPLADE | ColBERT | DPR | ANCE | GenQ | TAS-B | Contriever | UnifieR | LexMAE |
|---|---|---|---|---|---|---|---|---|---|---|---|
| In-Domain | 22.5 | 33.8 | 43.3 | 42.5 | - | 38.8 | 40.8 | 40.8 | - | 47.1 | **48.0** |
| TREC-COVID | 65.6 | 71.3 | 71.0 | 73.8 | 33.2 | 65.4 | 61.9 | 48.1 | 59.6 | 71.5 | **76.3** |
| NFCorpus | 32.5 | 32.8 | 33.4 | 33.8 | 18.9 | 23.7 | 31.9 | 31.9 | 32.8 | 32.9 | **34.7** |
| NQ | 32.9 | 39.9 | 52.1 | **56.2** | 47.4 | 44.6 | 35.8 | 46.3 | 49.8 | 51.4 | **56.2** |
| HotpotQA | 60.3 | 58.0 | 68.4 | 66.7 | 39.1 | 45.6 | 53.4 | 58.4 | 63.8 | 66.1 | **71.6** |
| FiQA | 23.6 | 29.1 | 33.6 | **35.6** | 11.2 | 29.5 | 30.8 | 30.0 | 32.9 | 31.1 | 35.2 |
| ArguAna | 31.5 | 34.9 | 47.9 | 46.3 | 17.5 | 41.5 | 49.3 | 42.9 | 44.6 | 39.0 | **50.0** |
| Tóuche-2020 | **36.7** | 34.7 | 27.2 | 26.3 | 13.1 | 24.0 | 18.2 | 16.2 | 23.0 | 30.2 | 29.0 |
| DBPedia | 31.3 | 33.1 | 43.5 | **44.6** | 26.3 | 28.1 | 32.8 | 38.4 | 41.3 | 40.6 | 42.4 |
| Scidocs | 15.8 | 16.2 | 15.8 | 15.4 | 7.7 | 12.2 | 14.3 | 14.9 | **16.5** | 15.0 | 15.9 |
| Fever | 75.3 | 71.4 | 78.6 | 78.5 | 56.2 | 66.9 | 66.9 | 70.0 | 75.8 | 69.6 | **80.0** |
| Climate-FEVER | 21.3 | 20.1 | **23.5** | 17.6 | 14.8 | 19.8 | 17.5 | 22.8 | 23.7 | 17.5 | 21.9 |
| SciFact | 66.5 | 67.5 | 69.3 | 69.3 | 31.8 | 50.7 | 64.4 | 64.3 | 67.7 | 68.6 | **71.7** |
| BEST ON | 1 | 0 | 1 | 3 | 0 | 0 | 0 | 0 | 1 | 0 | **7** |
| AVERAGE | 41.1 | 42.4 | 47.0 | 47.0 | 26.4 | 37.7 | 39.8 | 40.4 | 44.3 | 44.5 | **48.7** |

**TREC Deep Learning 2019 & 2020.** As shown in Table 1, we also evaluate our LexMAE on both the TREC Deep Learning 2019 (DL'19) and the TREC Deep Learning 2020 (DL'20). It is observed that LexMAE consistently achieves new state-of-the-art performance on both datasets.

**BEIR benchmark.** Meantime, we evaluate the LexMAE on BEIR benchmark, which contains twelve datasets, where ArguAna, Scidocs, Fever, Climate-FEVER, and SciFact are semantic relatedness tasks while TREC-COVID, NFCorpus, NQ, HotpotQA, FiQA, Tóuche-2020, and DBPedia are relevance-based retrieval tasks. To apply LexMAE pre-training to this benchmark, we pre-train LexMAE on BEIR collections and then fine-tune the pre-trained encoder on the in-domain supervised data of the BEIR benchmark. Lastly, we evaluate the fine-tuned LexMAE on both the in-domain evaluation set and the twelve out-of-domain datasets, whose results are listed in Table 2. It is observed that our LexMAE achieves the best in-domain performance. When performing the zero-shot transfer on the twelve out-of-domain datasets, LexMAE achieves the best on 7 out of 12 datasets, and delivers the best overall metrics, i.e., 'BEST ON' and 'AVERAGE', verifying LexMAE's generalization.

**MS-Marco Doc.** Lastly, we evaluate document retrieval on MS-Marco doc in Table 3: we pre-train LexMAE on the document collection and follow the fine-tuning pipeline of (Ma et al., 2022) (w/o distillation), where our setting is firstP of 384 tokens.

Table 3: Document Retrieval on Marco Doc Dev.

| Method | M@100 | R@100 |
|---|---|---|
| BERT | 38.9 | 87.7 |
| ICT (Lee et al., 2019) | 39.6 | 88.2 |
| B-PROP (Ma et al., 2021) | 39.5 | 88.3 |
| SEED (Lu et al., 2021) | 39.6 | 90.2 |
| COSTA (Ma et al., 2022) | 42.2 | 91.9 |
| LexMAE | **44.4** | **92.5** |

### 4.2 EFFICIENCY ANALYSIS AND COMPARISON

Here, we show efficacy-efficiency correlations after applying our top-K sparsifying (please see 3.4). A key metrics of retrieval systems is the efficiency in terms of retrieval latency (query-per-second, QPS), index size for inverted indexing, as well as representation size per document (for non-inverted indexing). On the one hand, as shown in Figure 2, our LexMAE achieves the best efficacy-efficiency trade-off among all dense-vector, quantized-dense, and lexicon-based methods. Compared to the previous state-of-the-art retriever, SimLM, we improve its retrieval effectiveness by 1.5% MRR@10 with $14.1\times$ acceleration. With top-L sparsifying, LexMAE can achieve competitive performance with SimLM with 100+ QPS. In addition, LexMAE shows a much better trade-off than the recent best PQ-IVF dense-vector retriever, RepCONC. Surprisingly, when only 4 tokens were kept for each passage, the performance of LexMAE (24.0% MRR@10) is still better than BM25 retrieval (18.5%).

On the other hand, as listed in Table 4, we also compare different retrieval paradigms in the aspect of their storage requirements. Note that each activated (non-zero) term in lexicon-weighed sparse vector needs 3 bytes (2 bytes for indexing and 1 byte for its quantized weight). Compared to dense-vector methods, lexicon-based methods, including our LexMAE, inherently show fewer storage requirements in terms of both index size of the collection and representation Byte per document.

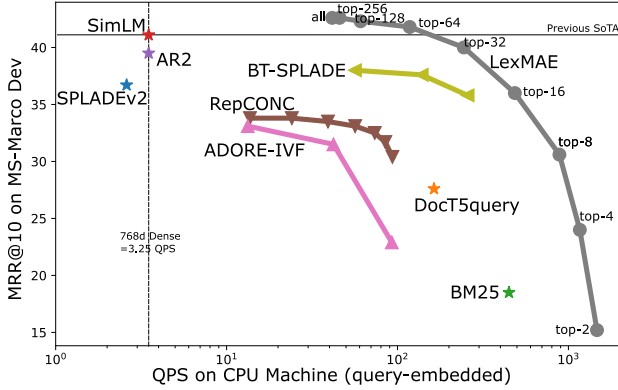

Figure 2: Retrievers w/ their MS-Marco dev MRR@10 and QPS, including dense-vector methods (i.e., SimLM, AR2), quantized-dense methods (i.e., RepCONC (Zhan et al., 2022), ADORE-IVF (Zhan et al., 2021)), and lexicon-based methods (i.e., SPLADEv2 (Formal et al., 2021a), BT-SPLADE (Lassance & Clinchant, 2022), DocT5query (Nogueira et al., 2019a), BM25, and ours).

| Method | IdxSize | ReprByte | M@10 |
|---|---|---|---|
| ColBERTv2 | 150G | 17,203 | 39.7 |
| AR2 | 27G | 3,072 | 39.5 |
| SimLM | 27G | 3,072 | 41.1 |
| BM25 | 4.3G | Avg 210 | 18.5 |
| SPLADE-max | 2.0G | Avg 290 | 34.0 |
| SPLADE-mask | 5.4G | Avg 915 | 37.3 |
| **LexMAE** | 3.7G | - | 42.6 |
| - top-256 sparsify | 3.5G | 768 | 42.6 |
| - top-128 sparsify | 2.4G | 384 | 42.3 |
| - top-64 sparsify | 1.4G | 192 | 41.8 |
| - top-32 sparsify | 0.9G | 96 | 40.0 |
| - top-16 sparsify | 0.5G | 48 | 36.0 |
| - top-8 sparsify | 0.4G | 24 | 30.6 |
| - top-4 sparsify | 0.3G | 12 | 24.0 |
| - top-2 sparsify | 0.2G | 6 | 15.2 |
| - top-1 sparsify | 0.2G | 3 | 1.9 |

Table 4: Index sizes (IdxSize) of models with retrieval performance (MRR@10) on MS-Marco Dev. 'ReprByte' denotes the storage requirement for an embedded passage.

Table 5: Ensemble & hybrid retrievers. [1]An ensemble of 4 SPLADE models.

| Method | M@10 | R@1 |
|---|---|---|
| **LexMAE**-pipeline | 43.1 | 28.8 |
| **LexMAE**-ensemble | 43.1 | 28.8 |
| UnifieR$_{uni}$ (Shen et al., 2022) | 40.7 | 26.9 |
| Ensemble of SPLADE[1] | 40.0 | - |
| COIL-full (Gao et al., 2021a) | 35.5 | - |
| CLEAR (Gao et al., 2021b) | 33.8 | - |

Table 6: Performance on different stages (see §A for their details) of the fine-tuning pipeline on MS-Marco Dev.

| Method | BM25 Negatives | | Hard Negatives | | Reranker-Distilled | |
|---|---|---|---|---|---|---|
| | M@10 | R@1k | M@10 | R@1k | M@10 | R@1k |
| coCondenser | 35.7 | 97.8 | 38.2 | 98.4 | 40.2 | 98.3 |
| SimLM | 38.0 | 98.3 | 39.1 | **98.6** | 41.1 | 98.7 |
| **LexMAE** | **39.3** | **98.3** | **40.8** | 98.5 | **42.6** | **98.8** |

Meantime, compared to BM25 building its index at the word level, the learnable lexicon-weighting methods, based on the smaller vocabulary of sub-words, are more memory-friendly.

## 4.3 Further Analysis

**Analysis of Dense-Lexicon Complement.** As verified by Shen et al. (2022), the lexicon-weighting retrieval is complementary to dense-vector retrieval and a simple linear combination of them can achieve excellent performance. As shown in Table 5, we conduct an experiment to complement our LexMAE with our re-implemented state-of-the-art dense-vector retrieval method, i.e., SimLM (Wang et al., 2022). Specifically, we propose to leverage two strategies: i) *ensemble*: a combination is applied to the retrieval scores of both paradigms, resulting in significant overheads due to twice large-scale retrieval; and ii) *pipeline*: a retrieval pipeline is leveraged to avoid twice retrieval: our lexicon-weighting retrieval is to retrieve top-K documents from the collection and then our dense-vector retrieval is merely applied to the constrained candidates for dense scores. It is shown that we improve the previous state-of-the-art hybrid retrieval method by 2.4% MRR@10 on MS-Marco Dev.

**Zero-shot Retrieval.** To figure out if our LexMAE pre-training can learn lexicon-importance distribution, we conduct a zero-shot retrieval on MS-Marco. Specifically, instead of $\mathrm{softmax}$ normalization function in Eq.(3), we present a saturation function-based L1 norm (i.e., $\mathrm{L1\text{-}Norm}(\log(1 + \mathrm{ReLU}(\cdot)))$) and keep the other parts unchanged. W/o fine-tuning, we apply the pre-trained LexMAE to MS-Marco retrieval task, with the sparse representation of $\log(1 + \mathrm{ReLU}(\cdot))$. As in Figure 3, our lexicon-importance embedding by LexMAE (110M parameters) beats BM25 in terms of large-scale retrieval and is competitive with a very large model, SGPT-CE (Muennighoff, 2022) with 6.1B parameters.

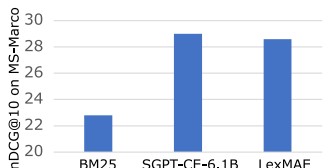

Figure 3: Zero-shot retrieval results (nDCG@10) on MS-Marco Dev.

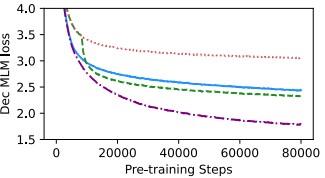 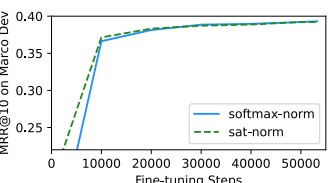

Figure 4: MLM pre-training losses (99% moving average has been applied) of LexMAE's encoder and decoder for various bottlenecks.

Figure 5: Fine-tuning MRR@10 curve on MS-Marco Dev.

**Multi-stage Retrieval Performance.** As shown in Table 6, we exhibit more details about the retrieval performance of different pre-training methods on multiple fine-tuning stages (see §A). It can be seen that our LexMAE consistently achieves the best or competitive results on all the 3 stages.

### 4.4 MODEL CHOICES & ABLATION STUDY

We conduct extensive experiments to check our model choices and their ablations from multiple aspects in Table 7. Note that, our LexMAE uses softmax-norm CBoW bottleneck, shares LM logits of encoder and bottleneck, and employs the inclusive masking strategy with 30% for the encoder and 50% for the decoder.

First, we try three other bottlenecks in Eq.(5), i.e., saturated norm for CBoW (as detailed in 'Zero-shot Retrieval'), dense bottleneck by using `[CLS]`, and no bottleneck by cutting the bottleneck off. As in Figure 4, their training loss curves show our CBoW bottlenecks do help the decoding compared to 'no-bottleneck', but are inferior to `[CLS]` contextual dense vector. But, attributed to pretraining-finetuning consistency, CBoW bottlenecks are better in lexicon-weighting retrieval. As for the two different lexicon CBoW bottlenecks, we show their fine-tuning dev curves in Figure 5: 'sat-norm' shows its great performance in early fine-tuning stages due to the same lexicon-representing way whereas 'softmax-norm' show better later fine-tuning results due to its generalization.

Table 7: First-stage (w/ BM25 negatives) fine-tuning results on MS-Marco Dev with different choices & ablations of LexMAE pre-training.

| Method | M@10 |
|---|---|
| **LexMAE** | 39.3 |
| *Bottleneck Choices* | |
| - Saturated Norm in CBoW (Eq.(3)) | 39.2 |
| - Dense `[CLS]` (Eq.(5)) | 38.6 |
| - Disable Bottleneck (Eq.(5)) | 38.5 |
| *Architecture Choices* | |
| - Enable Grad to $W^{(we)}$ in Bottleneck | 38.9 |
| - Share Encoder-Decoder LM Heads | 39.1 |
| - Add extra LM for Bottleneck | 39.2 |
| - DistilBERT-Initialized | 38.4 |
| *Masking Strategy ($\tilde{x}$ in Eq.(5))* | |
| - Exclusive | 39.1 |
| - Fully Random | 39.2 |
| *Masking Proportion (§3.1 & §3.3)* | |
| - Enc (15%) v.s. Dec (15%) | 38.4 |
| - Enc (15%) v.s. Dec (30%) | 38.8 |
| - Enc (30%) v.s. Dec (30%) | 39.0 |
| - Enc (40%) v.s. Dec (60%) | 39.0 |
| - Enc (30%) v.s. Dec (100%) | 39.0 |

Then, we make some subtle architecture changes to LexMAE: i) enabling gradient back-propagation to the word embedding matrix leads to a learning short-cut thus worse fine-tuning results; ii) both sharing the LM heads of our encoder and decoder (Eq.(1) and Eq.(5)) and adding an extra LM head specially for bottleneck LM logits (Eq.(3)) result in a minor drop, and iii) replacing BERT with DistilBERT (Sanh et al., 2019) for our initialization still outperforms a bunch of competitors.

Lastly, masking strategies other than our used 'inclusive' strategy in §3.3 do have minor effects on downstream fine-tuning. And, masking proportion of the encoder and decoder can affect the LexMAE's capability, and the negative 'affect' becomes unnoticeable when their proportion is large. In summary, pre-training LexMAE is very stable against various changes and consistently delivers great results. And please refer to §E for more experimental comparisons.

## 5 CONCLUSION

In this work, we propose to improve the lexicon-weighing retrieval by pre-training a lexicon-bottlenecked masked autoencoder (LexMAE) which alleviates the objective mismatch between the masked language modeling encoders and relevance-oriented lexicon importance. After pre-training LexMAE on large-scale collections, we first observe great zero-shot performance. Then after fine-tuning the LexMAE on the large-scale retrieval benchmark, we obtain state-of-the-art retrieval quality with very high efficiency and also deliver state-of-the-art zero-shot transfer performance on BEIR benchmark. Further detailed analyses on the efficacy-efficiency trade-off in terms of retrieval latency and storage memory also verify the superiority of our fine-tuned LexMAE.

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

## A  MULTI-STAGE RETRIEVER FINE-TUNING

### A.1  FINE-TUNING PIPELINE

To train a state-of-the-art lexicon-weighting retriever, we adapt the fine-tuning pipeline in a recent dense-vector retrieval method (Wang et al., 2022) as illustrated in Figure 6. The major difference is the source of the involved reranker for knowledge distillation into a retriever: In contrast to (Wang et al., 2022) that trains a reranker on the fly for retriever-specific reranker but suffers from high computation overheads, we propose to leverage an off-the-shelf reranker by (Zhou et al., 2022b).

**Stage 1: BM25 Negatives.**  In the first stage, we sample negatives for each query $q$ within top-$K_1$ document candidates by BM25 retrieval system, which is denoted as $\mathbb{N}^{(\text{bm25})}$. Therefore, the contrastive learning loss of stage 1 of our retriever fine-tuning is written as

$$L^{(\text{r1})} = \sum_q -\log P(\text{d} = d_+ | q, \{d_+\} \cup \mathbb{N}^{(\text{bm25})}; \theta^{(\text{enc})}) + \lambda_1 \text{ FLOPS}. \tag{11}$$

**Stage 2: Hard Negatives.**  Then, we sample the hard negatives $\mathbb{N}^{(\text{hn1})}$ for each query $q$ within top-$K_2$ candidates based on the relevance scores by the retriever we obtain in stage 1, and the training loss of our stage 2 is defined as

$$L^{(\text{r2})} = \sum_q -\log P(\text{d} = d_+ | q, \{d_+\} \cup \mathbb{N}^{(\text{hn1})}; \theta^{(\text{enc})}) + \lambda_2 \text{ FLOPS}. \tag{12}$$

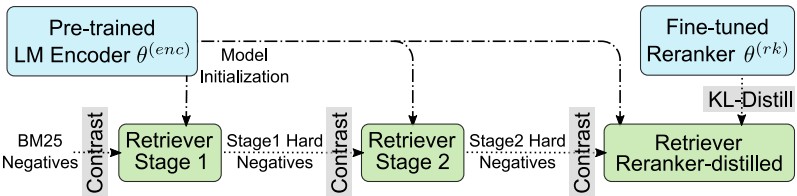

Figure 6: An illustration of the fine-tuning pipeline of our retrievers. Here, the fine-tuned reranker is directly adopted from (Zhou et al., 2022b) to avoid expensive reranker training.

**Stage 3: Reranker-Distilled.** Lastly, we further sample hard negatives $\mathbb{N}^{(\text{hn2})}$ for each query $q$ within top-$K_3$ candidates by the 2nd-stage retriever. Besides the contrastive learning objective, we also distill a well-trained reranker into our stage 3 of the retriever, which is written as

$$L^{(\text{r3})} = \sum_q \text{KL-Div}(P(\text{d}|q, \{d_+\} \cup \mathbb{N}^{(\text{hn2})}; \theta^{(\text{enc})}) || P(\text{d}|q, \{d_+\} \cup \mathbb{N}^{(\text{hn2})}; \theta^{(\text{rk})})) \qquad (13)$$
$$- \gamma \log P(\text{d} = d_+|q, \{d_+\} \cup \mathbb{N}^{(\text{hn2})}; \theta^{(\text{enc})}) + \lambda_3 \, \text{FLOPS} \,.$$

Here, $\theta^{(\text{rk})}$ parameterizes an expensive but effective cross-encoder as the reranker for knowledge distillation, and KL divergence, $\text{KL-Div}(\cdot||\cdot)$, is used as distillation loss with $\theta^{(\text{rk})}$ frozen.

### A.2 FINE-TUNING SETUPS

We share some hyperparameters across all three stages: learning rate is set to $2 \times 10^{-5}$ by following Shen et al. (2022), the number of training epochs is set to 3, model initialization is always our LexMAE, max document length is set to 144, and max query length is set to 32. Following (Wang et al., 2022), the $\gamma$ in Eq.(13) is set to 0.2. In contrast to (Wang et al., 2022) using 4 GPUs for fine-tuning, we limited all the fine-tuning experiments on one A100 GPU. The batch size (w.r.t the number of queries) is set to 24 with 1 positive and 15 negative documents in the fine-tuning stage 1 and 2, whereas that is set to 16 with 1 positive and 23 negative documents (to increase the number of negatives but fit one GPU memory by reducing the batch size). Another important hyperparameter is the depth (keeping how many top candidates as negatives) of negative sampling, i.e., $K$ in Eq.(11 - 13). By following Wang et al. (2022) and Gao & Callan (2022), we keep 1000 for BM25 negatives and 200 for other negatives, i.e., $K_1 = 1000, K_2 = 200, K_3 = 200$. The only hyperparameter we have tuned is the loss weight $\lambda$ in Eq.(11 - 13), i.e., $\lambda_1 \in \{0.001, 0.002, 0.004\}$ (corresponding to BM25 negatives) and $\lambda_{2/3} \in \{0.004, 0.008, 0.016\}$ (corresponding to hard negatives). Empirically, we found $\lambda_1 = 0.002, \lambda_2 = 0.008, \lambda_3 = 0.008$ achieving a great performance-efficiency trade-off. Again, we fix the random seed always to 42 in all our experiments without any tuning.

## B LEXICON-WEIGHTING INFERENCE FOR LARGE-SCALE RETRIEVAL

In the inference phase of large-scale retrieval, there are some differences between dense-vector and lexicon-weighting retrieval methods.

As in Eq.(10), we use the dot-product between the real-valved sparse lexicon-weighting representations as a relevance metric, where 'real-valved' is a prerequisite of gradient back-propagation and end-to-end learning. However, it is inefficient and infeasible to leverage the real-valved sparse representations, especially for the open-source term-based retrieval systems, e.g., LUCENE and Anserini (Yang et al., 2017). Following Formal et al. (2021a), we adopt 'quantization' and 'term-based system' to complete our retrieval procedure. That is, to transfer the high-dimensional sparse vectors back to the corresponding lexicons and their virtual frequencies, the lexicons are first obtained by keeping the non-zero elements in a high-dim sparse vector, and each virtual frequency then is derived from a straightforward quantization (i.e., $\lfloor 100 \times \boldsymbol{v} \rfloor$).

In summary, the overall procedure of our large-scale retrieval based on a fine-tuned LexMAE is i) generating the high-dim sparse vector for each document and transferring it to lexicons and frequencies, ii) building a term-based inverted index via Anserini (Yang et al., 2017) for all documents in a collection, iii) given a test query, generating the lexicons and frequencies, in the same way, and iv) querying the built index to get top document candidates.

## C    EXPLANATION OF DIFFERENT RECALL METRICS

Regarding R@N metric, we found there are two kinds of calculating ways, and we strictly follow the official evaluation one at `https://github.com/usnistgov/trec_eval` and `https://github.com/castorini/anserini`, which is defined as

$$\text{Marco-Recall@N} = \frac{1}{|\mathbb{Q}|} \sum_{q \in \mathbb{Q}} \frac{\sum_{d_+ \in \mathbb{D}_+} \mathbf{1}_{d_+ \in \bar{\mathbb{D}}}}{\min(N, |\mathbb{D}_+|)}, \tag{14}$$

where there may be multiple positive documents $\mathbb{D}^+ \in \mathbb{D}$, $\mathbb{Q}$ denotes the test queries and $\bar{\mathbb{D}}$ denotes top-K document candidates by a retrieval system. We also call this metric *all-positive-macro* Recall@N. On the other hand, another recall calculation method following DPR (Karpukhin et al., 2020) is defined as

$$\text{DPR-Recall@N} = \frac{1}{|\mathcal{Q}|} \sum_{q \in \mathbb{Q}} \mathbf{1}_{\exists d \in \bar{\mathbb{D}} \wedge d \in \mathbb{D}^+}. \tag{15}$$

which we call *one-positive-enough* Recall@N. Therefore, The official (*all-positive-macro*) Recall@N is usually less than DPR (*one-positive-enough*) Recall@N, and the smaller N, the more obvious.

Therefore, we make the unofficial *one-positive-enough* Recall@N standalone in Table 8 for more precise comparisons. It is observed that our LexMAE is still the best retriever among its competitors.

## D    SPARSIFYING LEXICON REPRESENTATIONS

Compared to dense-vector retrieval methods (Zhan et al., 2022; Xiao et al., 2022) that rely on product-quantization (PQ) and inverted file (IVF) to compromise their performance ($-3\% \sim 4\%$) for memory & time efficiency, the lexicon-weighting method with high-dimension sparse representation is intrinsically efficient for large-scale retrieval as demonstrated in §B – fully compatible with traditional term-based retrieval system, e.g., BM25 – only manipulating the term frequency and document frequency by the neural language modeling encoder.

To dive into LexMAE's efficacy-efficiency trade-off, we need to adjust the sparsity of lexicon representations for the documents. In general, the 'sparsity' here denotes how many lexicons in the vocabulary we used to represent each document. Since the hyperparameter $\lambda$ in Eq.(10) denotes the strength of sparse regularization during fine-tuning and controls the efficacy-efficiency trade-off for the retriever, it is straightforward to tune the $\lambda$ for the purpose of sparsifying. However, this requires fine-tuning the retriever multiple times with different $\lambda$ for adequate data points, leading to huge computation overheads. What's worse, there is no certain correlation between the $\lambda$ and the sparsity, leading to uncontrollable sparsifying and increasing the number of trials. To make the sparsifying procedure more controllable, Yang et al. (2021) and Lassance & Clinchant (2022) propose to sparsify the lexicon-weighing representations by controlling fine-tuning hyperparameters, e.g., constraining the number of activated lexicons, however still leading to extra fine-tuning efforts.

Therefore, in this work we present a simple but effective and controllable sparsifying method, which only presents during embedding documents in the inference phase and requires almost zero extra overheads. Specifically, it only keeps top-K weighted lexicons in the sparse lexicon representations $\boldsymbol{v}^{(d)} \in \mathbb{R}*^{|\mathbb{V}|}$ by Eq.(8), while removing the others by assigning zero weights, which can be formally written as

$$\hat{\boldsymbol{v}}_K^{(d)} = \boldsymbol{v}^{(d)} \odot \mathbf{1}_{\boldsymbol{v}^{(d)} \geq t}, \tag{16}$$

where $\odot$ denotes element-wise product, $t$ denotes the K-th large value in $\boldsymbol{v}^{(d)}$, and $\mathbf{1} \in \{0, 1\}^{|\mathbb{V}|}$ is a mask where an entry equals 1 only if the corresponding value in top-K of $\boldsymbol{v}^{(d)}$. If K is smaller than the number of activated (i.e., the weight $v_i^{(d)} > 0$) lexicons in $\boldsymbol{v}^{(d)}$, applying this sparsifying method would not make any change. All the sparsified lexicon representations $\hat{\boldsymbol{v}}_K^{(d)}$ with different K values are derived from the same original representation $\boldsymbol{v}^{(d)}$, so both the fine-tuning and embedding procedures are invoked only once, saving a lot of computing resources. Lastly, the sparsified lexicon representations, $\hat{\boldsymbol{v}}_K^{(d)}$, are used to build the inverted index for large-scale retrieval.

Table 8: Retrieval results on MS-Marco Dev on *one-positive-enough* recall. Note that the official Recall@50 of our fine-tuned LexMAE is 88.9%.

| Method | M@10 | *R@50* | *R@1K* |
|---|---|---|---|
| RocketQA (Qu et al., 2021) | 37.0 | 85.5 | 97.9 |
| PAIR (Ren et al., 2021a) | 37.9 | 86.4 | 98.2 |
| RocketQAv2 (Ren et al., 2021b) | 38.8 | 86.2 | 98.1 |
| AR2 (Zhang et al., 2022) | 39.5 | 87.8 | 98.6 |
| UnifieR$_{\text{lexicon}}$ (Shen et al., 2022) | 39.7 | 87.6 | 98.2 |
| UnifieR$_{\text{dense}}$ (Shen et al., 2022) | 38.8 | 86.3 | 97.8 |
| **LexMAE** | **42.6** | **89.6** | **99.0** |

Table 9: Comparisons of our retriever with *retrieval&rerank* pipelines.

| Retriever | Reranker | M@10 |
|---|---|---|
| RepBERT | RepBERT | 37.7 |
| ME-HYBRID | ME-HYBRID | 39.4 |
| RocketQA | RocketQA | 40.9 |
| RocketQAv2 | RocketQAv2 | 41.9 |
| **LexMAE** (retriever-only) | | 42.6 |

Table 10: Different pre-training objectives with its first-stage fine-tuning MRR@10 performance on MS-Marco.

| | **LexMAE** | SimLM | Enc-Dec MLM | Condenser | MLM | Enc-Dec RTD | AutoEncoder | BERT$_{\text{base}}$ |
|---|---|---|---|---|---|---|---|---|
| **M@10** | **39.3** | 38.0 | 37.7 | 36.9 | 36.7 | 36.2 | 32.8 | 33.7 |

# E  MORE EXPERIMENTS

**Comparison to Retrieval & Rerank Pipeline.**  Furthermore, our LexMAE retriever can outperform many state-of-the-art *retrieval & rerank* pipeline methods as in Table 9. It is noteworthy that these rerankers (a.k.a., cross-attention model or cross-encoder) are very costly as they must be applied to every query-document text concatenation, instead of query-agnostic representations from a bi-encoder.

**Comparisons with Different Pre-training Objectives.**  As listed in Table 10, we compare our pre-training objective with a batch of other objectives by fine-tuning the pre-trained model on MS-Marco BM25 negatives. It can be seen that our LexMAE improves the previous best by 1.3% MRR@10 in MS-Marco Dev.

# F  BACKGROUND: DENSE-VECTOR AND LEXICON-WEIGHTING RETRIEVAL

With recent surging pre-trained language models (PLMs) by self-supervised learning, e.g., GPT (Brown et al., 2020), BERT (Devlin et al., 2019), RoBERTa (Liu et al., 2019), and T5 (Raffel et al., 2020), deep representation learning has entered a new era to offer more expressively powerful text representations. As a practical task that relies heavily on text representation learning, large-scale retrieval directly benefits from these PLMs by leveraging the PLMs as neural encoders and fine-tuning them on the downstream datasets. Thereby, recent works upon the PLMs propose to learn encoders for large-scale retrieval, which are coarsely grouped into two paradigms according to their encoding spaces, i.e., dense-vector and lexicon-weighting retrieval.

**Dense-vector Encoding Methods.**  The most straightforward way to leverage the Transformer-based PLMs is directly representing a document/query as a fixed-length real-valued low-dimension dense vector $u \in \mathbb{R}^e$. By following the previous common practice, the dense vector is derived from either a contextual embedding of the special token '`[CLS]`' or a mean pooling over the sequence of word-level contextual embeddings. It is noteworthy that $e$ is embedding size and usually small (e.g., 768 for base-size PLMs), and the 'fixed-length' is not limited to one vector for each collection entry, but maybe multi-vector (Humeau et al., 2020; Khattab & Zaharia, 2020). Lastly, the relevance score between a document and a query is calculated by a very lightweight metric, e.g., dot-product or cosine similarity (Khattab & Zaharia, 2020; Xiong et al., 2021; Zhan et al., 2021; Gao & Callan, 2022). Although PLM-based dense-vector retrieval methods enjoy their off-the-shelf dense embeddings and easy-to-calculate relevance metrics, the methods are limited by their intrinsic representing manners – i) real-valued vectors leading to large index size and high retrieval latency and ii) high-level vector representations losing the key relevance feature about lexicon overlap.

**Lexicon-weighing Encoding Methods.** To make the best of word-level contextualization by considering either high concurrence (Nogueira et al., 2019a) or coordinate terms (Formal et al., 2021b) in PLMs, lexicon-weighing encoding methods, aligning more closely with the retrieval tasks, encode a query/document as a weighted sparse representation in vocabulary space (Formal et al., 2021b; Shen et al., 2022). It first weights all vocabulary lexicons for each word of a document/query based on the contexts, leading to a high-dimension sparse vector $v \in \mathbb{R}^{|\mathbb{V}|}$ ($|\mathbb{V}|$ is the vocabulary size and usually large, e.g., 30k). The text is then denoted by aggregating over all the lexicons in a sparse manner. More specifically, built upon causal language models (CLM) (Brown et al., 2020; Raffel et al., 2020), (Nogueira et al., 2019a) proposes to leverage the concurrence between a document and a query for lexicon-based sparse representation expansion. Built upon masked language models (MLM) (Devlin et al., 2019; Liu et al., 2019), Formal et al. (2021b) propose to couple the original word with top coordinate terms (full of synonyms and concepts) from the pre-trained MLM head. Lastly, the relevance is calculated by lexical-based matching metrics between the sparse lexicon representations (e.g., sparse dot-product and BM25 (Robertson & Zaragoza, 2009)).

