# OpenReview forum: "LexMAE: Lexicon-Bottlenecked Pretraining for Large-Scale Retrieval"
_ICLR.cc/2023/Conference — ICLR 2023 poster_

### Official Review · Reviewer_Bg1k · 2022-10-23

**Confidence:** 4
**Correctness:** 4
**Technical Novelty And Significance:** 4
**Empirical Novelty And Significance:** 4
**Recommendation:** 8

**Clarity, Quality, Novelty And Reproducibility:**

Clarity: Writing is clear and easy to follow

Novelty: Proposed pre-training framework is novel

Reproducibility: No code provided, hence no data point to verify the reproducibility


**Strength And Weaknesses:**

*Strength*
- Strong empirical results
- Controllable index size and low latency
- Predictions are explainable (advantage of sparse retrieval in general)

*Weakness*
- The sparse lexical presentation from encoder has a different form in pre-training (Eq. 3) and fine-tuning (Eq. 8). Why such disparity? Does performance drop if using the same?
- Many hyper-parameters to tune. For example, three-stage fine-tuning, alpha, and lambda, etc.


**Summary Of The Paper:**

This paper aimed to learn sparse representation for large-scale retrieval, which often enjoys low latency and smaller index size. The author proposed lexicon bottlenecked masked autoencoder (LexMAE). The encoder outputs an importance-aware lexicon representation and the decoder aims to reconstruct the missing tokens via the help of the lexicon representation. After three-staged fine-tuning, LexMAE not only achieved new SOTA results on MS-MARCO and TREC-DL benchmarks, but also enjoyed smaller index size and lower latency.

**Summary Of The Review:**

This is a novel work with strong empirical performance, which may have a large impact on the large-scale retrieval community. I would like to learn more details of LexMAE and its ablation.

Specifically,
- How scalable is LexMAE to a larger vocab space? In some applications (e.g., Product Search),  to index a wide range of model serial numbers or code-name of products, we may want the vocab space of size millions or more. Also, real-world text data can also be multilingual, hence the #sub-words can also be more than hundred of thousands.
- Does the pre-trained only (un-supervised) LexMAE always perform better than BM-25? For Table 1&2, can you include the Recall@k numbers of pre-trained only LexMAE, without fine-tuning on any supervised data (in-domain as well for BEIR)?
- How important is FLOP() term of Eq(10) in fine-tuning and how much it affect the performance?

---

> ### Author Response · Authors · 2022-11-11
> **Reply to Reviewer Bg1k (3/3)**
>
> - > **Question 4**: Does the pre-trained only (un-supervised) LexMAE always perform better than BM-25? For Table 1&2, can you include the Recall@k numbers of pre-trained only LexMAE, without fine-tuning on any supervised data (in-domain as well for BEIR)?
>
>     Thanks for your suggestions!
>
>     First of all, it is noteworthy that BM25 is a very strong baseline in zero-shot scenarios since it is general enough for relevance-oriented tasks including semantic relatedness and relevance-based retrieval tasks. As shown in [Zhou et al., 2022a], even a carefully designed retriever trained on pseudo labels with sophisticated objectives cannot beat a BM25 retrieval system.
>
>     Then, following your suggestions, we calculate the zero-shot metrics of LexMAE on both MS-Marco Passage and BEIR. As shown in the below tables, our pre-trained LexMAE without fine-tuning can surpass BM25 on Marco, NFCorpus, FiQA, and Climate-FEVER but performs worse in the remaining:
>
>     |                  | nDCG@10 | Recall@100 |
>     |------------------|---------|------------|
>     | BM25             | 22.5    | 58.5       |
>     | Zero-shot LexMAE | 28.6    | 21.5       |
>
>     | nDCG@10          | Marco | TREC-COVID | NFCorpus | NQ   | HotpotQA | FiQA | ArguAna | Touche | DBPedia | Scidocs | Fever | Climate | SciFact |
>     |------------------|:------|------------|----------|:-----|:---------|:-----|:--------|:-------|:--------|:--------|:------|:--------|:--------|
>     | BM25             | 22.5  | 65.6       | 32.5     | 32.9 | 60.3     | 23.6 | 31.5    | 36.7   | 31.3    | 15.8    | 75.3  | 21.3    | 66.5    |
>     | Zero-shot LexMAE | 28.6  | 28.6       | 34.4     | 27.1 | 13.2     | 26.1 | 10.2    | 12.3   | 23.3    | 14.4    | 10.0  | 29.9    | 4.5     |
>
>
>     The most potential reason why zero-shot LexMAE performs not competitively across the datasets is that contrastive learning objectives based on pseudo data are absent in LexMAE for relevance-aware representation learning. Such pseudo data heuristically collected upon in-domain corpora and corpus-aware contrastive learning objectives have been proven very effective in zero-shot retrieval by many previous works [Gao & Callan, 2022; Zhou et al., 2022a]. These suggest our LexMAE has great potential in zero-shot retrieval, which we leave in future work.
>
>
> - > **Question 5**: How important is FLOPS() term of Eq(10) in fine-tuning and how much it affect the performance?
>
>     The strength of the FLOPS trades off the lexicon-weighting retriever between effectiveness and efficiency. Consistent with the prior works [Formal et al., 2021a; Formal et al., 2022], we found that the loss weight of FLOPS (i.e., the strength of sparse regularization) is negatively correlated with the retriever's performance while positively with the representation sparsity (thus retrieval efficiency). Meantime, the FLOPS regularization term must be present during fine-tuning because an adequate sparsity of lexicon representations is the prerequisite for building an inverted index and conducting term-based retrieval.
>
>     In our work, to reduce the fine-tuning overheads during seeking a sweet point between effectiveness and efficiency, we heuristically tuned the loss weight lambda of FLOPS among a very small set of moderate-strength candidates and then proposed to sparsify the lexicon representation only during inference (see Figure 2, Table 4, and Appendix D).
>
>
>
> - **SUPPLEMENTAL REFERENCES**:
>
>     [Zhang et al., 2021] Mr. TyDi: A Multi-lingual Benchmark for Dense Retrieval, Workshop on Multilingual Representation Learning, EMNLP'21, https://arxiv.org/abs/2108.08787
>
>     [Xue et al., 2021] mT5: A massively multilingual pre-trained text-to-text transformer, NAACL'21, https://aclanthology.org/2021.naacl-main.41/

---

> ### Author Response · Authors · 2022-11-11
> **Reply to Reviewer Bg1k (2/3)**
>
> - > **Question 3**: How scalable is LexMAE to a larger vocab space? In some applications (e.g., Product Search), to index a wide range of model, serial numbers or code-name of products, we may want the vocab space of size millions or more. Also, real-world text data can also be multilingual, hence the #sub-words can also be more than hundred of thousands.
>
>     Thanks for these insightful questions!
>
>     For the scalability of the vocab size in LexMAE, we think it's adequate to leverage a moderate-size subword vocabulary for the monolingual retrieval, and the reasons are two-fold: i) The subword tokenizer, intrinsically acting like word lemmatizing or word stemming, is a great language analyzer to enable fuzzy matching, reduce vocabulary mismatch, preserve sparsity of the lexicon representations, and boost the semantic search. ii) A moderate-size vocabulary can save both GPU memory consumption and computation overheads, making large-scale pre-training and fine-tuning of LexMAE more practically feasible.
>
>     For the retrieval applications with a large body of rare words/strings (e.g., product names, serial numbers, code-name of products), we think this is an intrinsic challenge for a wide spectrum of semantic retrievers. Although the tokenized subwords in LexMAE can preserve partial information of the rare words (e.g., without ordering and proximity), a more feasible remedy is to complement the semantic matching-oriented LexMAE with a string overlap-oriented BM25 retrieval system that contains a large vocabulary with the rare words and strings.
>
>
>     Next, there are two scenarios of multilingual retrieval: i) zero-shot cross-lingual transfer (see the benchmark Mr. TyDi [Zhang et al., 2021]) which aims to train a retriever in a high-resource source language (e.g., English) and then transfer the retriever to a low-/zero-resource target language (e.g., Thai), and ii) multilingual retrieval which mixes multilingual documents into one unified collection and retrieves the relevant documents given a query in an arbitrary language. Here, the second scenario is more practical because it's closer to the commercial search engine.
>
>
>     As there is no previous work applying lexicon-weighting retrieval (e.g., SPLADE) to multilingual scenarios, recently we've conducted some pilot experiments on inhouse data for the second scenario and had the following observations: i) Directly fine-tuning a lexicon-weighting model (e.g., SPLADE) on multilingual retrieval cannot even reach convergence, and we found this is caused by an "accident translation" problem [Xue et al., 2021]. That is, a lexicon from the input will be accidentally translated into other languages, which is toxic for the multilingual retrieval task. ii) When applying LexMAE to this multilingual scenario, the fine-tuned model can finally achieve convergence, but the "accident translation" problem cannot be fully solved, resulting in an inferior performance. iii) To one extreme, we constrained the outputs into the lexicons from the input document/query -- degrading the model for just reweighting (and even compression) without any expansions. The empirical results show that such an input-constrained lexicon-weighting model surpasses the BM25 baseline by a large margin (68% v.s. 27% MRR@10). Also, it performs competitively with a fine-tuned dense-vector retriever (69% MRR@10) with 4000x faster retrieval speed (136 v.s. 0.03 QPS on our 100M multilingual collection). These observations suggest that lexicon-weighting retrieval and the LexMAE on multilingual are under-explored and deserve future research efforts.

---

> ### Author Response · Authors · 2022-11-11
> **Reply to Reviewer Bg1k (1/3)**
>
> Thanks for your in-depth comments! We sincerely hope our replies could address your concerns.
>
> - > **Question 1**: The sparse lexical representation from the encoder has a different form in pre-training (Eq. 3) and fine-tuning (Eq. 8). Why such disparity? Does performance drop if using the same?
>
>     We leverage a more general formulation to derive the lexicon-importance distribution in Eq.(3) to demonstrate our pre-training is not constrained to a single normalization function, while we fix the log-saturation function during fine-tuning as in Eq.(8) because it has been proven effective in previous lexicon-weighting retrieval papers [Formal et al., 2021a, Formal et al., 2022].
>
>     As shown in the experiments of our model choices (Table 7), our main model with softmax-norm (as stated below Eq.(3)) in pre-training delivers 39.3% MRR@10 whereas the model with saturated-norm in pre-training reaches 39.2% MRR@10, which verified that i) the generality of softmax-norm brings a little better performance and ii) our LexMAE pre-training is stable to different normalization functions. Meantime, benefiting from the pretraining-finetuning consistency, the pre-trained model with the saturated norm achieves better zero-shot performance and faster convergence (see Figure 5).
>
> - > **Question 2**: Many hyper-parameters to tune. For example, three-stage fine-tuning, alpha, lambda, etc.
>
>     Most of our hyperparameters followed the previous state-of-the-art paper [Wang et al., 2022] without tuning, including the three-stage fine-tuning, encoder masking ratio (alpha), decoder masking strategy, decoder masking ratio (beta), etc.
>
>     Just one hyperparameter we need to tune is lambda -- the loss weight of the sparse regularization term, FLOPS, in Eq.(10). As verified by previous lexicon-weighting papers [Formal et al., 2021a, Formal et al., 2022], the lambda is highly correlated to the sparsity of a lexicon representation, thus affecting the index size and retrieval latency. Please see our reply to your "Question 5" for more details about the FLOPS.

---

### Official Review · Reviewer_12su · 2022-10-24

**Confidence:** 4
**Correctness:** 4
**Technical Novelty And Significance:** 3
**Empirical Novelty And Significance:** 3
**Recommendation:** 8

**Clarity, Quality, Novelty And Reproducibility:**

I think the paper is well written and the motivation is very good.
The paper is with high quality.
The implementation of the approach should be pretty straightforward, given a researcher with background in MAE and large-scale retrieval system.

**Strength And Weaknesses:**

The paper is well written and the motivation is very clear. I love the idea of mapping the text into lexicon space and uses the inverted index as retrieval method to achieve better performance and better efficiency.
It is also great to see the improved performance for passage retrieval on MSMarco and better efficiency comparing with the dense embedding.

I have some quick questions:
1. Conceptually speaking, why would the sparse embedding (lexicon representation) approach achieves better performance than dense representation model?
2. For Figure 2, how to calculate the query per second for the dense embedding approach? I just curious whether the sparse embedding approach achieves better efficiency than the dense representation, especially given the fact that there also exists retrieval system for dense representation (e.g. faiss[1] etc.) Are you arguing that the sparse representation is more efficient than the dense one on the CPU?
3. For appendix D, is it sparsify instead of specify?
4. For appendix D, I wonder what is the top-K referring to? Given a text, we can encode it into a sparse embedding. Does the top-K mean the top-K activations on the sparse embedding?
5. I just curious what is the performance if not using the continuous bag-of-word technique?


[1]Johnson, Jeff, Matthijs Douze, and Hervé Jégou. "Billion-scale similarity search with gpus." IEEE Transactions on Big Data 7.3 (2019): 535-547.

**Summary Of The Paper:**

This paper proposed a new training framework for a language model that represents text in a lexicon space. The main motivation is that the large-scale retrieval (first stage retrieval) requires a high retrieval performance and a low latency. The dense representation approach could not work smoothly with the indexing technique developed in the search engine (inverted index etc.) Therefore, a method that maps the text in a sparse latent space is required. To learn such model, the paper proposed an encoder-decoder style self-supervised training framework. The encoder first encodes the text as a sparse embedding. However as the sparse embedding spans over the whole vocabulary. It is trivial to decode the full text from the sparse embedding. To add the difficulties, the author proposes the encode the sparse embedding into a continuous bag-of-word vector. The bag-of-word vector is concatenated with the masked text to be the input for the decoder. The proposed approach achieves a better performance than other state-of-the-art methods. It also achieves faster inference speed (in terms of the query per second metrics.)

**Summary Of The Review:**

I think the paper is novel and interesting to read. The observation of efficiency-efficacy is pretty interesting and important for real-world product. I think this paper is a good one and should be accepted.

---

> ### Author Response · Authors · 2022-11-11
> **Reply to Reviewer 12su (2/2)**
>
> - > **Question 4**: For appendix D, I wonder what is the top-K referring to? Given a text, we can encode it into a sparse embedding. Does the top-K mean the top-K activations on the sparse embedding?
>
>     Yes, the "Top-K" in Section 4.2 and Appendix D means that during document embedding in the inference phase, we only keep the activated lexicons with top-K weights on the sparse embeddings derived from Eq.(8) to represent the document while discarding the remaining.
>
>     For example, after we get a sparse lexicon embedding,
>
>     *{"manhattan": 202, "amid": 170, "project": 154, "communication": 151, "presence": 149, ... }* (corresponding to the passage with the id of 0 in the MS-Marco passage collection),
>
>     "top-2 sparsifying" means that we merely use *{"manhattan": 202, "amid": 170"}* to denote the passage. Sorry for such a confusing description of "top-K" in Appendix D and we've made it clearer in the revision.
>
> - > **Question 5**: I am just curious what is the performance if not using the continuous bag-of-word technique?
>
>     Thanks for this great question! It's intractable to directly feed such a high-dimensional lexicon-importance distribution from Eq.(3) into the Transformer-based decoder. This initially motivated us to propose the continuous bag-of-word (CBoW) bottleneck in this work.
>
>     Furthermore, a plausible strategy to remove CBoW is selecting top-weighted lexicons upon the lexicon-importance distribution, which can compose a discrete prompt to support the decoding procedure. As such, this pre-training potentially benefits not only the relevance-oriented tasks but also key words generation tasks. Nonetheless, such a "selection" action cannot be learned via back-propagation and thus poses optimization challenges (require reinforcement learning or the Gumbel trick), which we leave in future work.
>
> - **SUPPLEMENTAL REFERENCES**:
>
>     [Hofstatter et al. 2020] Improving Efficient Neural Ranking Models with Cross-Architecture Knowledge Distillation, ArXiv'2020, https://arxiv.org/abs/2010.02666

---

> ### Author Response · Authors · 2022-11-11
> **Reply to Reviewer 12su (1/2)**
>
> Thanks for your insightful comments! We will try our best to address your concerns and answer your questions.
>
> - > **Question 1**: Conceptually speaking, why would the sparse embedding (lexicon representation) approach achieve better performance than the dense representation model?
>
>     As first-stage retrieval is a task of calculating relevance between a query and a collection of documents, "lexicon overlap" is a crucial feature of relevance [Zhou et al., 2022a], which can be preserved and even expanded by lexicon representations. Therefore, lexicon-weighting models (e.g., SPLADE and LexMAE), as fully-learnable expansions of BM25, has great potential to surpass dense-vector methods, which has been proven across zero-shot and fully-finetuned scenarios:
>
>     In a zero-shot scenario, as verified by [Zhou et al., 2022a], a totally unlearned BM25 retrieval method can substantially outperform a sophisticated pseudo-supervised pre-trained dense-vector retrieval model (+9.0% R@20 on MS-Marco).
>
>     In a fully-finetuned scenario, a distilBERT-based dense-vector retriever can only achieve a 32.3 MRR@10 [Hofstatter et al. 2020]. In contrast, a lexicon-weighting method, SPLADEv2, also based on the DistilBERT model, can deliver 34.0 MRR@10 [Formal et al., 2021a] (i.e., 1.7% performance gap).
>
> - > **Question 2**: For Figure 2, how to calculate the query per second for the dense embedding approach? I am just curious whether the sparse embedding approach achieves better efficiency than the dense representation, especially given the fact that there also exists a retrieval system for dense representation (e.g. faiss[1] etc.) Are you arguing that the sparse representation is more efficient than the dense one on the CPU?
>
>     Sorry for missing these details.
>
>     To calculate the query per second (QPS) for the dense embedding approach, we conducted KNN-search for all the dev queries directly by the FAISS package on the same CPU machine and then calculated the QPS metric. For the lexicon representations, we directly leveraged the ANSERINI package to calculate the QPS.
>
>     Yes, we do argue that the sparse representation is more efficient than the dense one on the CPU machine. This is because the sparsity property of lexicon representation inherently enables the inverted index, which can largely improve retrieval efficiency. In contrast, dense-vector retrieval based on KNN search must calculate the relevance score for every query-document pair, thus requiring huge computation overheads. Furthermore, these dense-vector methods could resort to ANN search techniques, e.g., product quantization (PQ) and inverted file index (IVF), to accelerate their retrieval process, however with the sacrifice of performance (please refer to ADORE-IVF and RepCONC in Figure 2). For example, even if applying the most sophisticated ANN techniques, a considerable performance drop is still observed (e.g., −3%~4% by [Xiao et al., 2022]).
>
>
> - > **Question 3**: For appendix D, is it "sparsify" instead of "specify"?
>
>     Thanks for pointing this out, and we've fixed this typo in the revision.

---

### Official Review · Reviewer_v7Be · 2022-10-25

**Confidence:** 4
**Correctness:** 3
**Technical Novelty And Significance:** 3
**Empirical Novelty And Significance:** 3
**Recommendation:** 6

**Clarity, Quality, Novelty And Reproducibility:**

The structure of this paper is clear, the writing quality is good, and the method flow and experimental results are clearly displayed through a number of charts, which is convenient for readers to understand. The overall quality of this paper is good, and the method proposed in this paper has good novelty.

**Strength And Weaknesses:**

Strength:
（1）The innovation of this paper is novel. This work proposes a lexicon-bottlenecked masked autoencoder. LexMAE encoder assigns higher importance scores to essential vocabulary lexicons of the masked document but lowers to trivial ones. It is used to solve the gap between language modeling and lexicon-weighting retrieval in the vocabulary-weighted paradigm.
（2）The method proposed in this paper has a good effect and has achieved improvement on multiple datasets.
（3）The experimental analysis of this paper is sufficient. In addition to verifying the effectiveness of the proposed method on multiple datasets, this paper also conducts efficiency analysis, Zero-shot Retrieval, Multi-stage Retrieval Performance, and other experiments to comprehensively verify the effectiveness of the method in this paper.

Weaknesses:
(1) The sparsity defined in this paper is lack description, so it is hard to understand the experiments in Figure 2 and Table 4.  The efficiency of the proposed model is represented by increasing the sparsity, so it is necessary to make it clear in section 3.4.

**Summary Of The Paper:**

The lexicon-weighting paradigm in large-scale retrieval has achieved certain results. However, the language modeling prefers certain or low-entropy words whereas the lexicon-weighting retrieval favors pivot or high-entropy words. In view of the above problems, a lexicon-bottlenecked masked autoencoder (LexMAE) is proposed to learn the vocabulary representation of importance perception. The lexicon-importance distribution is learned unsupervised by constructing a continuous word package bottleneck. The experimental results show the effectiveness of the proposed method.

**Summary Of The Review:**

The writing structure of this paper is clear and the innovation is relatively new. The experiment has fully verified the performance of the method from many aspects, which is a good article on the whole.

---

> ### Author Response · Authors · 2022-11-11
> **Reply to Reviewer v7Be**
>
> We appreciate your constructive comments, and we will update the manuscript accordingly.
>
> - > **Question 1**: The sparsity defined in this paper lacks description, so it is hard to understand the experiments in Figure 2 and Table 4. The efficiency of the proposed model is represented by increasing the sparsity, so it is necessary to make it clear in section 3.4.
>
>     We appreciate for pointing this out.
>
>     To dive into LexMAE's efficacy-efficiency trade-off, we need to adjust the sparsity of lexicon representations for the documents. In general, the "sparsity" here denotes how many lexicons in the vocabulary we used to represent each document. Since the hyperparameter lambda in Eq.(10) denotes the strength of sparse regularization during fine-tuning and controls the efficacy-efficiency trade-off for the retriever, it is straightforward to tune the $\lambda$ for the purpose of sparsifying. However, this requires fine-tuning the retriever multiple times with different lambda for adequate data points, leading to huge computation overheads. What's worse, there is no predictable correlation between the lambda and the sparsity, leading to uncontrollable sparsifying and increasing the number of trials. To make the sparsifying procedure more controllable, [Yang et al., 2021] and [Lassance & Clinchant, 2022] propose to sparsify the lexicon-weighing representations by controlling fine-tuning hyperparameters, e.g., constraining the number of activated lexicons, however still leading to extra fine-tuning efforts.
>
>     Therefore, in this work we present a simple but effective and controllable sparsifying method, which only presents during embedding documents in the inference phase and requires almost zero extra overheads. Specifically, it only keeps top-K weighted lexicons in the sparse lexicon representations v^{(d)} \in \R*^{|\sV|} by Eq.(8), while removing the others by assigning zero weights. If K is smaller than the number of activated (i.e., the weight v^{(d)}_i > 0) lexicons in v^{(d)}, applying this sparsifying method would not make any change. All the sparsified lexicon representations with different K values are derived from the same original representation v^{(d)}, so both the fine-tuning and embedding procedures are invoked only once, saving a lot of computing resources. Lastly, the sparsified lexicon representations are used to build the inverted index for large-scale retrieval.
>
>     Due to the page limit, we merged such descriptions about sparsifying to Appendix D (please see the revision). Meantime, please see an example of the sparsifying as follows. Given a passage from the MS-Marco collection, we can encode it to produce its sparse lexicon representations, v^{(d)}, which can be written in a dictionary format: *{"manhattan": 202, "amid": 170, "project": 154, "communication": 151, "presence": 149, ... }* (corresponding to the passage with the id of 0 in the MS-Marco passage collection), where the keys denote lexicons in the vocabulary, their values denote the corresponding weights in v^{(d)}, and the items are sorted according to the values in descending order. Here, "top-2 sparsifying" means that we merely use *{"manhattan": 202, "amid": 170"}* to denote the passage.

---

> > ### Comment · Reviewer_v7Be · 2022-11-28
> > **Thanks**
> >
> > The author's responses address most of my concerns and I suggest adding some detailed descriptions in the main context.

---

> > > ### Author Response · Authors · 2022-11-30
> > > **Thank Reviewer v7Be for the reply**
> > >
> > > Thanks for your reply, and we're pleased to address your concerns. Following your suggestion, we will add a new short paragraph in the main context to briefly explain the top-K sparsifying. Due to the closure of revision, we put the paragraph here and will add it to Section 3.4 of the manuscript:
> > >
> > > > **Top-K Sparsifying.** Attributed to inherent flexibility, we can adjust the sparsity of the lexicon-weighting representations for the documents to achieve a targeted efficacy-efficiency trade-off. Here, the "sparsity" denotes how many lexicons in the vocabulary we used to represent each document. Previous methods either tune sparse regularization strength (Formal et al., 2021a; 2022) (e.g., $\lambda$ in Eq.(10)) or propose other sparse hyperparameters (Yang et al., 2021; Lassance & Clinchant, 2022) (e.g., the number of activated lexicons), however causing heavy fine-tuning overheads. Hence, we present a simple but effective sparsifying method, which only presents during embedding documents in the inference phase so requires almost zero extra overheads. It only keeps top-K weighted lexicons v(d)  in the representations $v^{(d)}\in R*^{|\mathcal{V}|}$ by Eq.(8), while removing the others by assigning zero weights (see Appendix D for details). We will dive into empirical efficacy-efficiency analyses later in Section 4.2.
> > >
> > > If you have any follow-up questions, please no hesitate to raise any. Many thanks again for your efforts in reviewing and improving our manuscript!

---

### Official Review · Reviewer_KtuL · 2022-11-03

**Confidence:** 3
**Correctness:** 3
**Technical Novelty And Significance:** 3
**Empirical Novelty And Significance:** 3
**Recommendation:** 6

**Clarity, Quality, Novelty And Reproducibility:**

Clarity: As I mentioned above. I think the paper should be quite clear to researchers who are familiar with the terminologies and background to text retrieval. However, it takes some efforts for others who are not directly working on the task to catch up.

Novelty: I do believe the paper has some nice technical contributions, especially on how to design the bottleneck and make the pre-training more friendly to lexicon-weighting based retrieval (though it is a bit too specific a task with the cost of pre-training).

Reproducibility: I don't seem to find a section that's dedicated to technical details or talking about code or model release. So I think it takes some efforts if one wants to reproduce the results presented in the paper.

**Strength And Weaknesses:**

Strengths:
- While I am less familiar with the detailed metrics used for text retrieval, the results presented in the paper is the most impressive part to me. dIt seems LexMAE can significantly improve prior-arts, and also maintain a very healthy speed-accuracy trade-off. The evaluation is also not just done on one dataset, but on multiple datasets.
- I also think the technical contributions of the paper is of good value. For example in how to construct a reasonable bottleneck to make the task meaningful enough to pre-train representations, while also get aligned with the downstream task (text retrieval with lexicon-weighting).

Weaknesses:
- One high-level weakness is that pre-training is really supposed to work with not just one task, but with multiple tasks. Right now the pre-training is adapted in way that works particularly well for one task. I am not saying this is bad, but doing pre-training for just one task sounds a bit less scalable -- for 1000 tasks it has to design 1000 pre-training models and actually run them.
- The writing of the paper is also a bit unfriendly to folks who are directly working on text retrieval. It mentioned about different ways of doing retrieval (dense or lexicon-weighting) at the beginning of the paper, and only explain such terms in more detail in later parts. It could be made better to just consolidate all the background knowledge into one section, for easier reference and make the paper more self-contained.

**Summary Of The Paper:**

The paper presents a pre-training method specifically for large-scale retrieval in the text domain. Specifically, it uses an auto-encoder architecture, with a bottleneck on the pooled lexicon-based representations in between the encoder and the decoder. After pre-training, the model is adapted to the task of large-scale text retrieval (with a technique called lexicon-weighting retriever). In terms of empirical studies, the paper benchmarked on several text retrieval tasks, either with full fine-tuning, or with zero-shot transfer. Also a lot of analyses are done, including a very extensive one for efficiency. According to the tables and figures, the proposed approach (LexMAE) achieves impressive improvement over methods compared.

**Summary Of The Review:**

Overall, I think the strengths outweighs the weaknesses for the paper. The paper does some task-specific pre-training, and shows some good results. The writing is probably okay for researchers within the domain but a bit effort-taking for ones outside of the domain. The results are impressive and the empirical validations are extensive, but there are concerns about the reproducibility of the paper.

---

> ### Author Response · Authors · 2022-11-11
> **Reply to Reviewer KtuL (2/2)**
>
> - > **Question 2**: The writing is probably okay for researchers within the domain but a bit effort-taking for ones outside of the domain.
>
>     Thanks for your constructive suggestion! Please see the below paragraphs about the background of dense-vector and lexicon-weighting retrieval. Meantime, to satisfy the requirement of the page number, we added these paragraphs into a "Background" section (see Appendix F in the revision) and will try to merge them into the main paper:
>
>     With recent surging pre-trained language models (PLMs) by self-supervised learning, e.g., GPT, BERT, RoBERTa, and T5, deep representation learning has entered a new era to offer more expressively powerful text representations. As a practical task that relies heavily on text representation learning, large-scale retrieval directly benefits from these PLMs by leveraging the PLMs as neural encoders and fine-tuning them on the downstream datasets. Thereby, recent works upon the PLMs propose to learn encoders for large-scale retrieval, which are coarsely grouped into two paradigms according to their encoding spaces, i.e., dense-vector and lexicon-weighting retrieval.
>
>     *Dense-vector Encoding Methods*: The most straightforward way to leverage the Transformerbased PLMs is directly representing a document/query as a fixed-length real-valued low-dimension dense vector u \in R^e. By following the previous common practice, the dense vector is derived from either a contextual embedding of the special token "[CLS]" or a mean pooling over the sequence of word-level contextual embeddings. It is noteworthy that e is embedding size and usually small (e.g., 768 for base-size PLMs), and the "fixed-length" is not limited to one vector for each collection entry, but maybe multi-vector. Lastly, the relevance score between a document and a query is calculated by a very lightweight metric, e.g., dot-product or cosine similarity. Although PLM-based dense-vector retrieval methods enjoy their off-the-shelf dense embeddings and easy-to-calculate relevance metrics, the methods are limited by their intrinsic representing manners i) real-valued vectors leading to large index size and high retrieval latency and ii) high-level vector representations losing the key relevance feature about lexicon overlap.
>
>
>     *Lexicon-weighing Encoding Methods*: To make the best of word-level contextualization by considering either high concurrence [Nogueira et al., 2019a] or coordinate terms [Formal et al., 2021b] in PLMs, lexicon-weighing encoding methods, aligning more closely with the retrieval tasks, encode a query/document as a weighted sparse representation in vocabulary space. It first weights all vocabulary lexicons for each word of a document/query based on the contexts, leading to a high-dimension sparse vector v \in R^{|V|} (|V| is the vocabulary size and usually large, e.g., 30k). The text is then denoted by aggregating over all the lexicons in a sparse manner. More specifically, built upon causal language models (CLM), [Nogueira et al., 2019a] propose to leverage the concurrence between a document and a query for lexicon-based sparse representation expansion. Built upon masked language models (MLM), [Formal et al., 2021b] propose to couple the original word with top coordinate terms (full of synonyms and concepts) from the pre-trained MLM head. Lastly, the relevance is calculated by lexicon-based matching metrics between the sparse lexicon representations (e.g., sparse dot-product and BM25).
>
>
> - > **Question 3**: but there are concerns about the reproducibility of the paper.
>
>     We will make our implemented codes, pre-trained models, and fine-tuned models publicly available after being de-anonymized.
>
>
> - **SUPPLEMENTAL REFERENCES**:
>
>     [Wei et al., 2022] Emergent Abilities of Large Language Models, TMLR'22, https://arxiv.org/abs/2206.07682
>     [Gururangan et al. 2020] Don't Stop Pretraining: Adapt Language Models to Domains and Tasks, ACL'20, https://aclanthology.org/2020.acl-main.740/

---

> ### Author Response · Authors · 2022-11-11
> **Reply to Reviewer KtuL (1/2)**
>
> Thanks for your in-depth comments. We will carefully revise our manuscript according to your suggestions.
>
> - > **Question 1**: One high-level weakness is that pre-training is really supposed to work with not just one task, but with multiple tasks. Right now the pre-training is adapted in way that works particularly well for one task. I am not saying this is bad, but doing pre-training for just one task sounds a bit less scalable -- for 1000 tasks it has to design 1000 pre-training models and actually run them.
>
>     We appreciate these in-depth comments!
>
>     After the community entered the "pre-training" era, recent improvements to adapt pre-trained models have been made along with two directions roughly. On the one hand, in line with the thought of "multiple tasks", some works, especially large language models (LLMs), unify a variety of task categories or/and formulations in a self-supervised (e.g., GPT-3, GLM, PaLM) or supervised (e.g., T5, FLAN, T0, InstructGPT) manner. These works benefit from world knowledge and facilitate knowledge transfer to a broad spectrum of downstream language tasks. However, a recent finding by [Wei et al., 2022] shows that these models have the emergent ability after scaling up to 100 billion parameters, which inevitably limits their practical deployments, especially in large-scale retrieval.
>
>     On the other hand, following the inspiration of "Don't Stop Pretraining" [Gururangan et al. 2020], a large body of works focus on adapting pre-trained language models to the targeted domains or tasks. They carefully design task-specific self-supervised objects on selective corpora to reduce the pretraining-finetuning inconsistency in terms of domains, formulations, etc., with small extra continual pre-training overheads. For example, LexMAE only costs about 1/10 extra GPU hours than its initialization, BERT. Although "for 1000 tasks it has to design 1000 pre-training models and actually run them", this paradigm could circumvent negative transfer and deliver very state-of-the-art performance on the targeted task while keeping the pre-trained model at a relatively small scale and preserving its practicality. In line with this paradigm, our proposed LexMAE bridges the gap between the general MLM objectives and the downstream relevance-oriented lexicon-weighting objectives by introducing a brand-new lexicon bottleneck, which is empirically proven effective in large-scale retrieval.

---

### Author Response · Authors · 2022-11-11
**About the Reproducibility of the LexMAE framework**

We deeply appreciate all reviewers for the constructive and in-depth comments.

To ensure the reproducibility of the proposed LexMAE framework, we will make our implemented codes, pre-trained models, and fine-tuned retrievers publicly available after being de-anonymized.

---

### Decision · Program_Chairs · 2023-01-20

**Decision:**

Accept: poster

**Justification For Why Not Higher Score:**

As stated in the meta review, when the proposed method is evaluated in pure zero-shot, the performance compared to BM25 is much less impressive. However, the comparison in Table 2 includes methods which do not use as much data for finetuning. I would like the authors to include this result and discuss that in the main paper.

**Justification For Why Not Lower Score:**

The paper is well motivated and the experiments support the claims. Reviewers have judged this work as well executed.

**Metareview: Summary, Strengths And Weaknesses:**

This paper introduces a lexicon-bottlenecked masked auto encoder (LexMAE) that allows to train text representations. The motivation behind this work is to obtain high throughput encoders that allow to index large text databases for document retrieval. The proposed method is well motivated and described, the experiments presented support the claims. It has been noted that while obtaining good results, the proposed approach offers a healthy speed accuracy tradeoff. As weaknesses, as noted by reviewer Bg1k, some of the evaluations, especially comparing with BM25 or contriver are a bit misleading because of the amount of data available to fine-tune the model. The authors have provided zero-shot evaluations, which should be included in the main paper. The results are less overwhelming that in the case of finetuning, but do not invalidate the merit of this work. Given all that is stated above, I recommend accepting this paper as a poster.

**Note From Pc:**

if the above contains the word "oral" or "spotlight" please see: "oral" presentation means -> notable-top-5% and "spotlight" means -> notable-top-25%. As stated in our emails, we are disassociating presentation type from AC recommendations